# Genetic and environmental determinants of variation in the plasma lipidome of older Australian twins

Matthew WK Wong[1], Anbupalam Thalamuthu[1], Nady Braidy[1], Karen A Mather[1,2], Yue Liu[1], Liliana Ciobanu[1,3], Bernhardt T Baune[3,4,5,6], Nicola J Armstrong[7], John Kwok[8], Peter Schofield[2,9], Margaret J Wright[10,11], David Ames[12,13], Russell Pickford[14], Teresa Lee[1,15], Anne Poljak[1,9,14†], Perminder S Sachdev[1,15†]*

[1]Centre for Healthy Brain Ageing, School of Psychiatry, Faculty of Medicine, University of New South Wales, Sydney, Australia; [2]Neuroscience Research Australia, Sydney, Australia; [3]The University of Adelaide, Adelaide Medical School, Discipline of Psychiatry, Adelaide, Australia; [4]Department of Psychiatry, University of Münster, Münster, Germany; [5]Department of Psychiatry, Melbourne Medical School, The University of Melbourne, Melbourne, Australia; [6]The Florey Institute of Neuroscience and Mental Health, The University of Melbourne, Melbourne, Australia; [7]Mathematics and Statistics, Murdoch University, Perth, Australia; [8]Brain and Mind Centre, The University of Sydney, Sydney, Australia; [9]School of Medical Sciences, University of New South Wales, Sydney, Australia; [10]Queensland Brain Institute, University of Queensland, Brisbane, Australia; [11]Centre for Advanced Imaging, University of Queensland, Brisbane, Australia; [12]University of Melbourne Academic Unit for Psychiatry of Old Age, Kew, Australia; [13]National Ageing Research Institute, Parkville, Australia; [14]Bioanalytical Mass Spectrometry Facility, University of New South Wales, Sydney, Australia; [15]Neuropsychiatric Institute, Euroa Centre, Prince of Wales Hospital, Sydney, Australia

*For correspondence:
p.sachdev@unsw.edu.au

†These authors contributed equally to this work

Competing interests: The authors declare that no competing interests exist.

**Abstract** The critical role of blood lipids in a broad range of health and disease states is well recognised but less explored is the interplay of genetics and environment within the broader blood lipidome. We examined heritability of the plasma lipidome among healthy older-aged twins (75 monozygotic/55 dizygotic pairs) enrolled in the Older Australian Twins Study (OATS) and explored corresponding gene expression and DNA methylation associations. 27/209 lipids (13.3%) detected by liquid chromatography-coupled mass spectrometry (LC-MS) were significantly heritable under the classical ACE twin model ($h^2$ = 0.28–0.59), which included ceramides (Cer) and triglycerides (TG). Relative to non-significantly heritable TGs, heritable TGs had a greater number of associations with gene transcripts, not directly associated with lipid metabolism, but with immune function, signalling and transcriptional regulation. Genome-wide average DNA methylation (GWAM) levels accounted for variability in some non-heritable lipids. We reveal a complex interplay of genetic and environmental influences on the ageing plasma lipidome.

## Introduction

As the field of lipidomics has grown, hundreds to thousands of complex lipids have been characterised (*Fahy et al., 2005*; *Quehenberger et al., 2010*), with many linked to health and disease states, such as metabolic syndrome (*Meikle and Christopher, 2011*), cardiovascular disease (*Meikle et al., 2014*; *Harmon et al., 2016*), obesity (*Barber et al., 2012*; *Rauschert et al., 2016*), and dementia

(*Han et al., 2001*; *Kim et al., 2018*; *Mielke et al., 2012*; *Wong et al., 2017*). Both genetic and environmental factors influence these biological phenotypes. Identifying the contributions of these factors can help elucidate the importance of genes for a particular trait, as well as providing insight into the environmental influences. This information might enable the design of personalised medical treatments for lipid-related disease states.

While there are substantial data to suggest that levels of traditional lipids and lipoproteins such as high density lipoprotein (HDL) cholesterol, low density lipoprotein (LDL), cholesterol and triglyceride levels are heritable (*Liu et al., 2018*; *Goode et al., 2007*), few studies have focused on the genetic and environmental influences on the plasma levels of individual lipid species and lipid classes beyond these traditional lipid measures. Additionally, lipids vary within and between individuals (*Begum et al., 2016*; *Begum et al., 2017*; *Saw et al., 2017*) based on variables such as age (*Ishikawa et al., 2014*; *Lawton et al., 2008*; *Wong et al., 2019a*), sex (*Ishikawa et al., 2014*; *Wong et al., 2019a*), body mass index (BMI) (*Wong et al., 2019a*; *Shamai et al., 2011*), lipid-lowering medication (*Meikle et al., 2015*) and genetic background (*Liu et al., 2018*; *Bennet et al., 2007*), demonstrating a wide degree of complexity involved in the regulation of lipid metabolism. It would therefore be informative to understand the extent to which variation in specific plasma lipids is determined by genetic and environmental influences. We hypothesise that as circulating lipids are produced downstream of genomic, transcriptomic and proteomic regulatory processes, that there will be strong environmental influences on lipid variance.

Previous genome-wide association study (GWAS) data implicate many genetic loci associated with traditional lipid levels. For example, the genes encoding lipoprotein lipase, hepatic lipase and cholesteryl ester transfer protein (*LPL, LIPC* and *CETP* respectively) have been associated with HDL, and genes encoding cadherin EGF LAG seven-pass G-type receptor 2, apolipoprotein B and translocase of outer mitochondrial membrane 40 (*CELSR2, APOB* and *TOMM40* respectively) have been associated with LDL (*Middelberg et al., 2011*). Apolipoprotein E (*APOE*) variants have been established as a strong risk factor for cardiovascular disease and Alzheimer's disease (*Bennet et al., 2007*; *Corder et al., 1994*) and are associated with altered LDL-C levels. One large exome wide screening study with over 300,000 individuals identified 444 variants at 250 loci to be associated with one or more of plasma LDL, HDL, total cholesterol and triglyceride levels (*Liu et al., 2017*). Collectively, data from 70 independent GWAS with sample sizes ranging from ten thousand to several hundred thousand participants have identified associations of traditional lipid levels with 500 single nucleotide polymorphism (SNPs) in 167 loci that explain up to 40% of individual variance in these traditional lipid measures (*Matey-Hernandez et al., 2018*). This number suggests that LDL, HDL, total cholesterol and triglyceride levels undergo a substantial degree of genetic regulation, but also highlights that much of the lipid variance is still unaccounted for, possibly related to rare variants or environmental factors (*Matey-Hernandez et al., 2018*; *García-Giustiniani and Stein, 2016*).

One of the most powerful tools for analysis of gene versus environment effects on phenotypic traits is the classical twin design, which estimates the relative contribution of heritable additive genetic effects (A) and shared (C) and unique environmental (E) influences on a given trait by comparing correlations within monozygotic and dizygotic twin pairs (*van Dongen et al., 2012*). One major strength of this design compared to family studies is that twins are matched by age and common environment, reducing cross-generation differences. Genetic and environmental variances can be computed with relatively high power using a modest sample size. It is expected that since monozygotic twins share 100% of segregating genetic variation, while dizygotic twins share 50%. It is also assumed that twins are raised in the same environment, thus any additional differences between monozygotic twins would be attributable to unique environmental (E) effects. Further, any differences in intraclass correlations between monozygotic and dizygotic twins could be estimated as due to additive polygenic effects (A).

We applied the classic twin design to estimate heritability using 75 pairs of MZ twins and 55 pairs of DZ twins from the Older Australian Twin Study (OATS) (*Sachdev et al., 2009*; *Sachdev et al., 2013*), aged between 69–93 years. Since many proteins are known to regulate lipid metabolism, it is expected that some lipids may show substantial heritability, as reported in previous studies (*Frahnow et al., 2017*; *Draisma et al., 2013*). Further, we hypothesised that some of the variance in lipids that do not have significant heritability might be controlled by gene sequence - independent mechanisms, such as genome-wide average DNA methylation (GWAM) levels. Our study is the first

to examine heritability of the broad plasma lipidome among healthy older – aged twins and explore putative genetic, transcriptomic and epigenetic associations of these lipids.

# Results

## Participant characteristics

Plasma lipidomics was performed on $n$ = 330 individuals, 260 of these were used for heritability analyses. Characteristics of the MZ ($n$ = 150, 100 females) and DZ ($n$ = 110, 79 females) twins with available plasma for heritability analyses are presented in *Table 1*. There were no group differences between MZ and DZ twins on these characteristics except in HDL-C levels, which were higher in MZ twins relative to DZ twins (p<0.05), but did not remain significant after correcting for multiple comparisons.

## Heritability

Heritability of lipids was computed using the classical ACE model. Classical lipid measures of total cholesterol, LDL, HDL and triglycerides were significantly heritable ($h^2$ = 0.427, 95% C.I. = [0.075, 0.592], 0.404, 95% C.I. = [0.121, 0.573], 0.419, 95% C.I. = [0.027, 0.766], and 0.427, 95% C.I. = [0.181, 0.623] respectively). HDL had a substantial C component (i.e., common environment; $h^2_C$ = 0.27, 95% C.I. = [0.00, 0.48]). For individual lipid species measured, 27 out of 203 (13.3%) were significantly heritable with a median heritability of $h^2$ = 0.433, ranging from 0.287 for TG (18:0/17:0/18:0) to a maximum of 0.59 for Cer (d17:1/24:1).

The percentages of heritable lipids from the total pool of identified lipids in each lipid class is summarised in *Figure 1A*. Heritability estimates across lipid class and by individual lipid for significantly heritable lipids are summarised in *Figure 1B* and *Supplementary file 1A*. Ceramides (Cer) had the highest heritability estimates (range $h^2$ = 0.433–0.59), where 9 out of 20 species were significantly heritable. For triglycerides (TG), 12 of out 59 species measured were heritable (range $h^2$ = 0.287–0.495). Among diacylglycerols (DG), 3 species out of 10 were heritable (range $h^2$ = 0.422–0.544). Only three phospholipids were heritable, including 2 of 58 phosphatidylcholines (PC) and 1 out of 5 phosphatidylethanolamines (PE), (range $h^2$ = 0.327–0.413). Cholesteryl ester (CE), lysophosphatidylcholine (LPC), phosphatidylinositol (PI) and SM (sphingomyelin) species were not significantly heritable, with median heritability for non-significant lipids at $h^2$ = 0.23, and near zero heritability for

**Table 1.** Participant characteristics for heritability analyses.

|  | MZ (n = 150) | DZ (n = 110) | Statistic | p-value |
|---|---|---|---|---|
| Age | 75.7 (5.47) | 76.07 (5.31) | −0.548 | 0.584 |
| Females | 100 (67%) | 79 (72%) | 0.785 | 0.376 |
| Education (yrs) | 10.99 (3.18) | 11.2 (3.18) | −0.475 | 0.635 |
| BMI (kg/m$^2$) | 27.934 (4.74) | 27.5 (4.92) | 0.776 | 0.438 |
| WHR | 0.89 (0.09) | 0.89 (0.08) | 0.164 | 0.87 |
| MMSE | 28.9 (1.37) | 28.95 (1.76) | −0.062 | 0.95 |
| LDL-C (mmol/L) | 2.77 (0.97) | 2.78 (0.97) | −0.078 | 0.938 |
| HDL-C (mmol/L) | 1.73 (0.46) | 1.60 (0.44) | 2.341 | 0.02 |
| Cholesterol (mmol/L) | 5.08 (1.01) | 4.98 (1.12) | 0.822 | 0.412 |
| Triglyceride (mmol/L) | 1.30 (0.54) | 1.32 (0.56) | −0.298 | 0.766 |
| *APOE* ε4 carrier* | 35 (26%) | 27 (28%) | 0.118 | 0.731 |

Means (SD) are presented for continuous variables, while n (%) is presented for categorical variables. Comparisons of MZ and DZ pairs used *t* tests for continuous variables and χ (**Quehenberger et al., 2010**) tests for categorical variables.

Abbreviations: MZ = monozygotic, DZ = dizygotic. body mass index (BMI), mini-mental state exam (MMSE), waist-hip ratio (WHR), low density lipoprotein cholesterol (LDL-C), high density lipoprotein cholesterol (HDL-C).

*excludes participants with missing data (n = 231 participants with *APOE* genotype data).

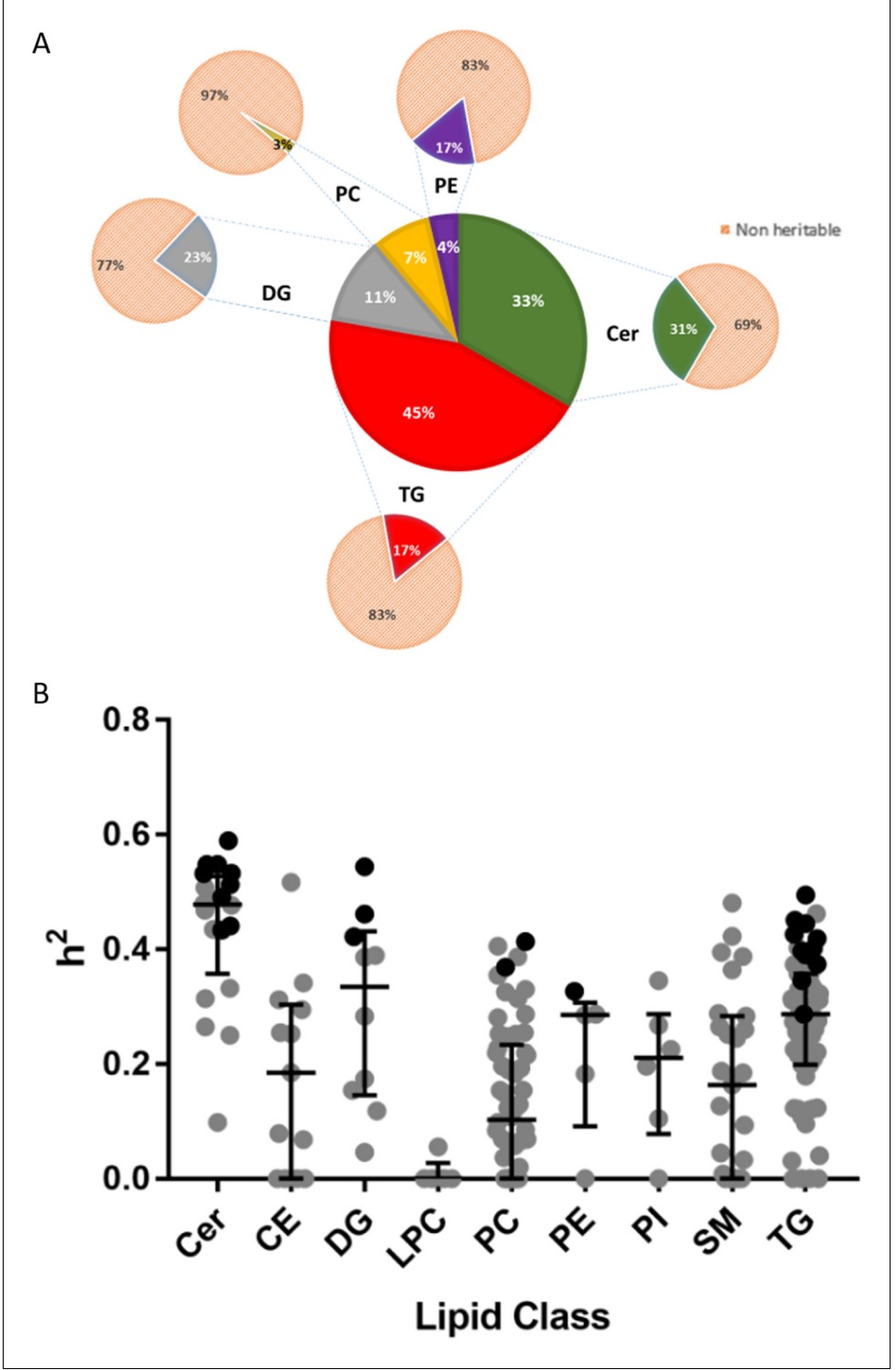

**Figure 1.** Heritability of lipids. (**A**) Percentage distribution of heritable lipids. The central wheel represents significantly heritable lipids and their percentage distribution by lipid class. Smaller wheels emanating from each sector represent proportions of these heritable lipids compared to total measured lipids of that class, such that the sum of these smaller wheels equals the total pool of 207 individual lipids measured. For example, 45% of significantly heritable lipids belonged to the TG lipid class, and these heritable lipids represented 17% of total measured plasma TG. Orange sectors represent non-heritable percentage of each lipid class. (**B**) The distribution of heritability ($h^2$), estimated from the ACE model, for each individual lipid species grouped according to class. Boxplots show median with interquartile range for each class. Dark circles represent heritable lipids, as opposed

*Figure 1 continued on next page*

*Figure 1 continued*

to grey circles, which represent lipids that were not significantly heritable. Minimum (significant) heritability is $h^2 > 0.287$.

The online version of this article includes the following figure supplement(s) for figure 1:

**Figure supplement 1.** Genetic correlation heatmap.

LPC species. Heritability estimates obtained for summed lipid groups (*Supplementary file 1B*) were mostly similar to that of the individual lipids, though there were some differences. For example, the sum of monounsaturated SM species was heritable whereas no individual SM was significantly heritable. A complete heritability table for all lipids is presented in *Supplementary file 2A*.

Additionally, a sex heterogeneity model was used to assess differences in heritability between sexes (*Supplementary file 2B*), while a gene-environment interaction model was used to assess heritability differences between age groups (*Supplementary file 2C*). We found suggestive levels of significance for four lipids between sexes (unadjusted $p < 0.05$), and a substantial effect of age on heritability estimates in both directions (decreasing with age, as well as increasing with age) across most significantly heritable lipids.

## Genetic, Environmental and Phenotypic Correlations

Genetic and environmental correlations were estimated for significantly heritable lipid species and lipid classes. Median genetic correlations between Cer species were high ($r_g = 0.94$), as were TG ($r_g = 0.81$) and DG ($r_g = 0.73$) species. DG and TG were also highly genetically correlated with each other ($r_g = 0.70$), as were Cer species with monounsaturated SM ($r_g = 0.83$). Median phenotypic correlations between Cer species, between TG species and between DG species were $r_p = 0.85, 0.61$, and $0.53$ respectively, and $r_p = 0.51$ between TG and DG species, and $r_p = 0.83$ between Cer and monounsaturated SM. Median unique environmental correlations were moderately lower than corresponding genetic correlations ($r_e = 0.75, 0.56$ and $0.53$ for Cer, TG and DG respectively, and $r_e = 0.45$ between TG and DG, and $r_e = 0.72$ between Cer and monounsaturated SM), indicating that heritable lipids of similar class have a strong shared genetic basis relative to the unique environment. Further, traditional lipids (LDL-C, HDL-C, total cholesterol and TG) had poor genetic and phenotypic correlations with individual lipid species, apart from traditional triglyceride measures, which was highly correlated with individual TG and DG species. A genetic correlation matrix heatmap is shown in *Figure 1—figure supplement 1*.

## Association with gene expression

The association of lipids (n = 209) with probe level gene expression (n = 35,971) was analysed using linear mixed models via the R package nlme (*Pinheiro et al., 2019*). We found significant gene expression probe associations (n = 3568) with 47 individual lipids (7 DG, 2 PC, 1 PE, 37 TG; see *Supplementary file 2D and 2E*). Of these associations, 15 were linked to significantly heritable lipids (12 TG, 3 DG, n = 380 unique probes). In fact, we found that all significantly heritable TG and DG species were also significantly associated with gene expression of particular transcripts. An additional 32 individual lipids (25 TGs, 4 DGs, 2 PCs and 1 PE, n = 276 unique probes) without significant heritability were significantly associated with gene expression. In regards to traditional and grouped classes of lipids, there were also significant gene expression associations with HDL-C, total TG, and grouped TGs regardless of total carbon number or number of double bonds. No significant gene expression associations were identified for LDL-C. There was a modest but non-significant positive correlation between variance explained by gene expression of probes and heritability ($p > 0.05$, *Figure 2* and *Supplementary file 2D*). This implies that gene expression accounts for some but not all the variance in heritable lipid levels.

Since the bulk of significant gene expression associations were with TG, we examined the relationship of gene expression associations for TG species by degree of saturation, classifying each TG species as being saturated (no fatty acyl double bonds), monounsaturated (possessing one double bond), or polyunsaturated (possessing two or more double bonds). We then investigated how many transcripts were associated with a low, medium and high number of lipids, by counting the number of gene transcripts significantly associated with either 1–2 lipids, 3–8 lipids, and over eight lipids in

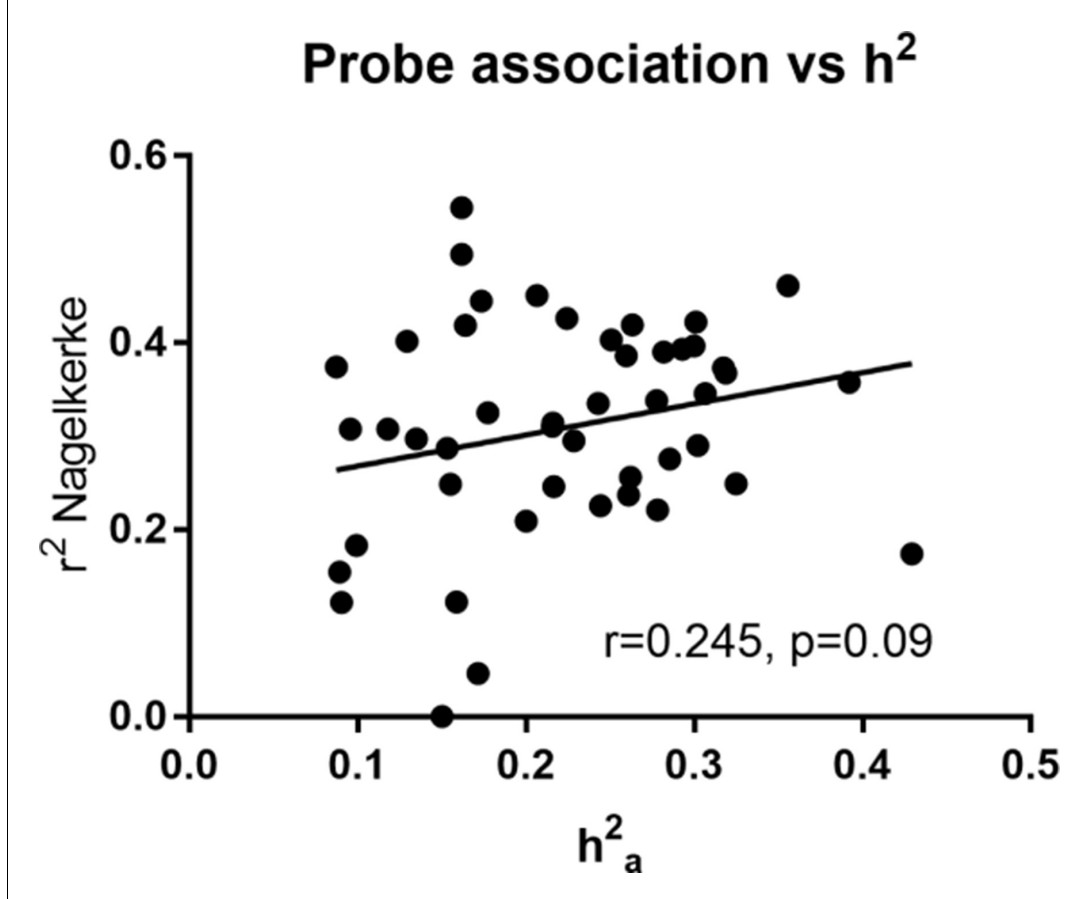

**Figure 2.** Heritability estimate ($h^2_a$) vs total variance explained (Nagelkerke $r^2$) by gene expression probe transcripts for heritable lipids. Pearson correlation was calculated.

The online version of this article includes the following figure supplement(s) for figure 2:

**Figure supplement 1.** Batch correction using inverse rank normal transform of residuals.

that class (in the case of polyunsaturated TG). Generally, only a few gene transcripts were associated with many lipids, regardless of saturation level. There were 282 gene transcripts associated with 1–2 TGs in the saturated TG class, but only six were associated with at least three different TGs in that class.

**Table 2.** Gene expression associations among TG lipids.

| TG class | Number of associated lipids | Number of transcript associations |
|---|---|---|
| Saturated TG | 1–2 | 282 |
| | 3–8 | 6 |
| Monounsaturated TG | 1–2 | 59 |
| | 3–8 | 7 |
| Polyunsaturated TG | 1–2 | 243 |
| | 3–8 | 119 |
| | >8 | 9 |

Note. Table lists number of gene expression associations common to a maximum of 1–2, 3–8 and >8 lipids in each TG saturation class (saturated, monounsaturated, and polyunsaturated TG).

*Table 2* summarises the number of significantly associated gene transcripts among each TG saturation class, while *Figure 3* is a Venn diagram identifying gene transcripts that are unique or shared across saturation classes for significantly heritable TG lipids (*Figure 3A*) and non-heritable TGs only (*Figure 3B*). The total list of gene transcripts associated with lipids can be found in *Supplementary file 2E and 2F*, while *Supplementary file 2G and 2H* show gene transcripts ordered by TG degree of saturation and total number of carbons. For example, ribosomal protein L4 pseudogene 2 (*RPL4P2*), A disintegrin and metalloproteinase domain-containing protein 8 (*ADAM8*) and Adipocyte Plasma Membrane Associated Protein (*APMAP*) were uniquely associated with saturated TG when considering heritable TG lipids.

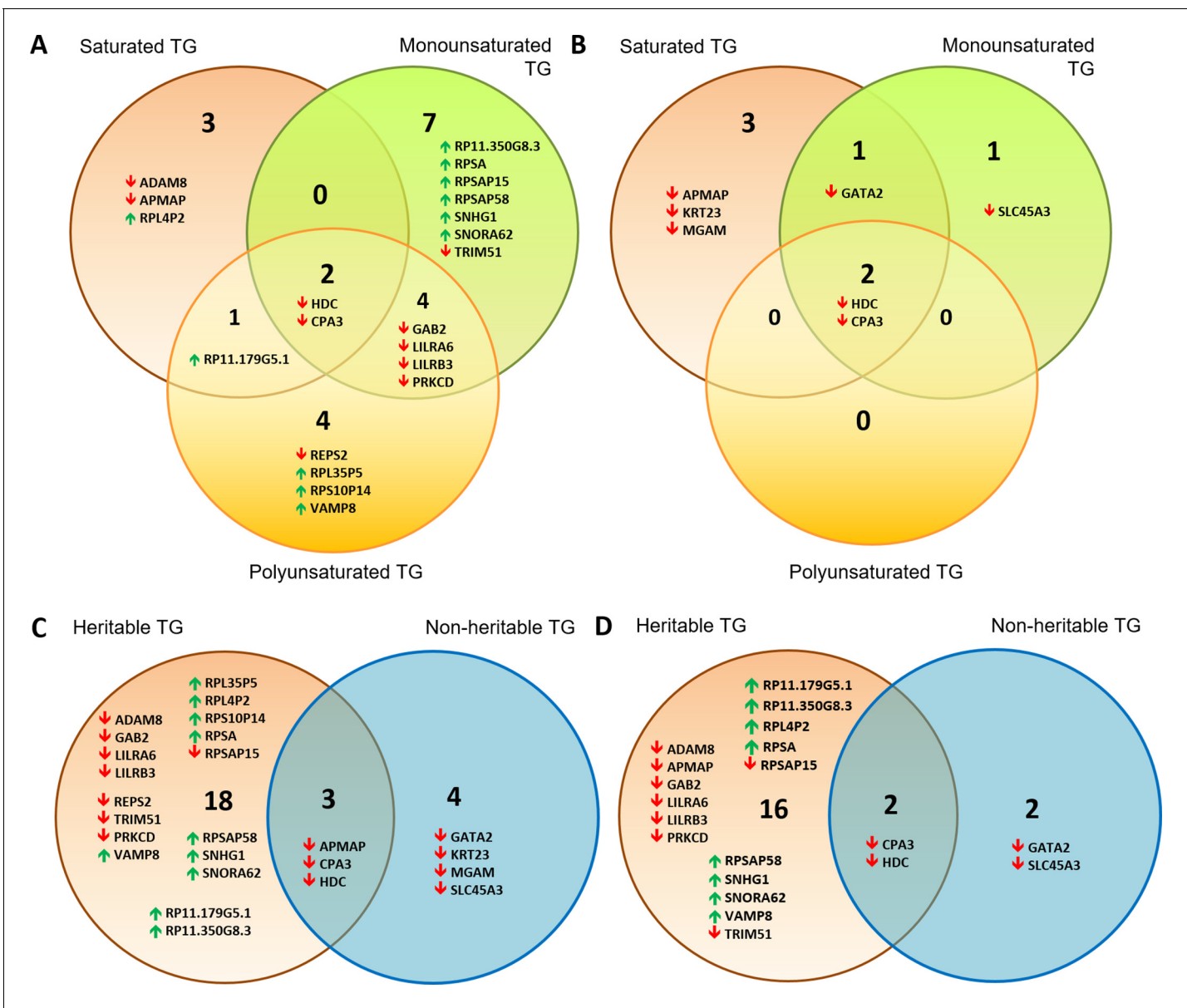

**Figure 3.** Venn diagrams showing distribution of gene transcripts associated with a majority of TG lipids. These were subdivided into those associated with saturated vs monounsaturated vs polyunsaturated lipids for (**A**) significantly heritable TGs and (**B**) non-heritable TGs. Also shown are heritable vs non-heritable set of significant gene expression associations of TG lipids that were first subdivided based on (**C**) double bond group/saturation (*Supplementary file 2G*) and (**D**) total number of carbons (<49 carbons, 49–55 carbons and 56+ carbons, *Supplementary file 2H*). Gene transcripts included in these Venn diagrams were those significantly associated with the highest and second highest number of lipids of a particular saturation class (**A and B**), or among heritable and non-heritable lipids (**C and D**). Upwards and downwards arrows indicate positive and inverse gene expression associations with lipid levels respectively.

Interestingly, there were a number of transcripts associated with a maximum of 1–2 TG lipids (e.g. 1–2 saturated lipids had 282 hits). In a majority of cases, these associations were driven by a specific TG lipid (among saturated TGs, this was TG(16_0/16_0/24_0), among monounsaturated TGs, this was TG(16_0/14_0/18_1) and for polyunsaturated TGs, these were TG(19_1/18_1/18_2), TG(16_0/18_1/23_1), TG(16_0/22_6/22_6) and TG(25_0/18_1/18_1)). These lipids tended to have a medium to high total carbon count (i.e. >55 carbons). By contrast, our analysis also found gene expression of histidine decarboxylase (*HDC*) and carboxypeptidase A3 (*CPA3*) to be significantly associated with all TGs irrespective of the number of total carbons and number of double bonds. In fact, *HDC* and *CPA3* were also significantly associated with other lipids including DG and HDL-C (*Supplementary file 2E*). Notably, there were some differences between the gene transcript association profiles of significantly heritable vs non-heritable lipids; many more gene transcript associations were unique to heritable as opposed to non-heritable TGs (*Figure 3A–D*, *Supplementary file 3*). For example, pseudogenes appearing in the heritable lipid list do not appear in the non-heritable list. Comparing transcribed genes associated with TG lipids by total number of carbons (<49 carbons 'low', 49–55 carbons 'medium' and 56+ carbons 'high') also yielded a similar outcome (*Figure 3D*).

Furthermore, the majority of transcriptome associations with non-heritable lipids were inverse associations, whereas the lipid-transcriptome associations for heritable lipids were a mix of positive and inverse associations, suggesting a diverse impact of these lipids on cellular function. It is also interesting that the majority of inverse lipid-transcriptome associations encode protein coding transcripts (15/17 total), and only 2/17 were non-protein coding RNAs/pseudogenes. By contrast, the majority of positive lipid-transcriptome associations were for non-protein coding pseudogenes (9/11) and only 2/11 were protein coding.

## Functional pathways of associated gene transcripts

We examined whether gene transcripts significantly associated with lipids included genes responsible for structural changes by a diversity of lipid modifying machinery, such as families of elongase and desaturase enzymes responsible for modifying fatty acid chain length and saturation level (*Kindt et al., 2018*), as well as a plethora of synthetases which assemble complex lipids such as the triglycerides and phospholipids (*Sorger and Daum, 2002*). Interestingly, in our gene/lipid transcriptomic association list (*Table 3*, *Figure 3* and *Figure 4*), such structure regulating genes do not appear.

Instead, the transcripts reveal genes which regulate other physiological and cellular functions particularly those involved with immune and vascular functions (*Table 3*), with possible roles in the central nervous system (CNS). We also found an upregulation of pseudogenes. The STRING and BioGRID databases (*Szklarczyk et al., 2019*; *Oughtred et al., 2019*) were used to provide functional information on genes identified in the lipid-transcriptome analysis. Some other notable pathways include vasoactive peptides, vesicular transport and pseudogenes/non-protein coding genes. The latter could play important regulatory roles, such as in gene silencing (*Guo et al., 2014*).

Directions of arrows indicate either positive (upwards facing) or inverse (downwards facing) lipid-gene transcriptome associations. Even though our transcriptomic data was for the blood transcriptome, some of these genes also have functions in the CNS or associations with neurodegenerative diseases (far right column).

## Association of DNA methylation levels at specific CpG sites with lipid and gene expression

To gain insight into the relationships between lipid levels and DNA methylation of CpGs at specific genes, we selected gene transcripts significantly associated with lipids and identified associations between DNA methylation at CpG sites within close proximity to these gene transcripts, and lipid expression (*Supplementary file 2I*). We found significant associations of DNA methylation ($p<0.05$) with four lipids: PE(16:0_20:4), TG(25:0_16:0_18:1), TG(18:0_17:0_18:0) and TG(18:1_18:2_18:2). Of these, two were heritable - TG(25:0_16:0_18:1) and TG(18:0_17:0_18:0).

We also examined the relationship between gene expression and DNA methylation at specific CpG sites of genes whose transcripts were associated with significant heritability (*Supplementary file 2J*). We found 19 significant CpG site-gene expression associations related to four unique lipids (TG(19:1_18:1_18:2), TG(15:0_16:0_18:1), PC(20:2_18:2), TG(16:0_18:1_23:1), but

**Table 3.** Functions of genes with significant lipid-gene transcriptome associations.

| Biological Pathways | Gene Transcripts* | Relevance to the CNS |
|---|---|---|
| Inflammation | | |
| Innate immunity | ↓LILRB3, ↓MGAM | |
| Adaptive immune response | ↓LILRA6 | |
| Host Defense | ↓FPR1, ↓TRIM51 | FPR1 found in neural glial cells, astrocytes and neuroblastoma (*Cussell et al., 2019*). |
| Allergic Response | ↓ADAM8, ↓HDC, ↓CPA3 | ADAM8 may regulate cell adhesion during neurodegeneration (*Schlomann et al., 2000*).<br>HDC as a histidine decarboxylase, produces histamine, which in the CNS is a neurotransmitter (*Yoshikawa et al., 2014*). |
| Class I MHC antigen binding | ↓LILRA6, ↓LILRB3 | |
| B-Cell response/receptor signalling | ↓GAB2, ↓LILRB3, ↓PRKCD | GAB2 is associated with Alzheimer's disease. By activating PI3K, increases amyloid production and microglia-mediated inflammation. Several *GAB2* SNPs are associated with late-onset Alzheimer's disease (*Chen et al., 2018*). |
| Mast Cell Degranulation | ↓CPA3, ↓HDC, | |
| Vasoactive Actions | | |
| Regulation of vasoactive peptides (e.g., endothelin, angiotensin 1, snake toxins, etc) | ↓GATA2, ↓CPA3, | |
| Epithelial Cell Integrity | ↓KRT23, ↓PRKCD | |
| Cell Adhesion | ↓APMAP | APMAP supresses brain Aβ production (*Mosser et al., 2015*). |
| DNA Regulation | ↑ RPSA, ↑ SNORA62, ↑ SNHG1 | |
| Vesicle/Endosome Regulation/Transport | ↑ VAMP8, ↓REPS2, ↓SLC45A3 | SLC45A3 regulates oligodendrocyte differentiation (*Shin et al., 2012*). |
| Pseudogenes/non-protein coding | ↓S100A11P1, ↓RPSAP15, ↑ RP11-179G5.1, ↑ RP11-350G8.3, ↑ RPL35P5, ↑ RPL4P2, ↑ RPS10P14, ↑ RPSAP15, ↑ RPSAP58, ↑ SNHG1, ↑ SNORA62 | Regulatory roles. Gene silencing, affects mRNA stability. |

these associations were a very minor subset of all CpG site-gene expression associations. Therefore, we did not find sufficient evidence to suggest that DNA methylation at specific CpG sites drives changes in gene expression, though we acknowledge this analysis lacks sufficient power to be conclusive.

## Association of lipids with genome wide average DNA methylation (GWAM)

We then explored associations of genome wide average methylation with lipid levels and found significant associations of all five LPCs (and the total LPC sum) with GWAM (range beta = −0.22 to −0.27, see *Table 4*). Notably, four TGs were also significantly inversely associated with GWAM (beta = −0.18 to −0.23). Further, only two other lipids were positively associated with GWAM, namely one CE and one PC (beta = 0.21, and 0.18 respectively, *Table 3*). None of these lipids was significantly heritable, with maximum heritability of 0.39, though one TG (TG18:1_17:1_22:6) was borderline significant (p=0.05 for h$^2$), with a maximum of two significant gene expression associations (for TG18:1_17:1_22:6 and TG18:1_20:4_22:6).

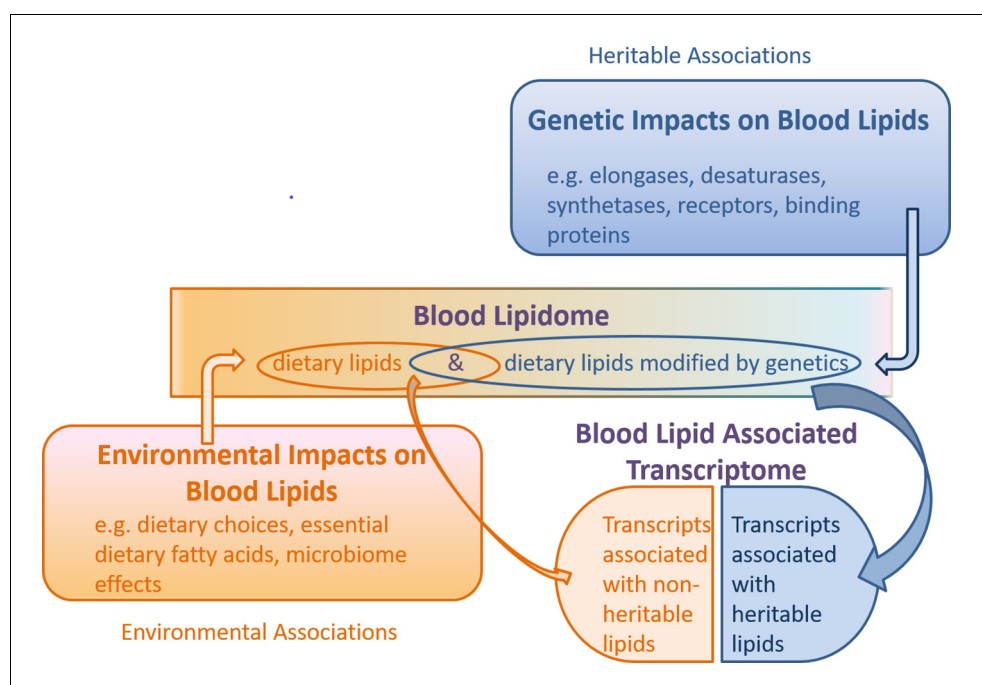

**Figure 4.** Schematic of the combined genetic and environmental influences on the blood lipidome, and the association of this lipidome with the blood transcriptome. Under this model, non-heritable lipids could affect gene transcription, while heritable lipids could also affect gene transcription (collectively 'blood lipid associated transcriptome'), but are possibly modified upstream by genetic machinery such as elongases, desaturases, synthetases, receptors and binding proteins. Gene transcripts encoding these enzymes and proteins may be independent of the 'blood lipid associated transcriptome' noted in this study.

## Discussion

In this study, we evaluated the relative contributions of genetic versus environmental factors to the plasma lipidome among older Australian twins aged 69–93 years. As hypothesised, both genetic and environmental factors contribute to shaping the plasma lipidome, though in our sample of older

**Table 4.** Regression of lipid residuals significantly associated with genome wide average DNA methylation levels.

| Lipid | Beta | SE | t | p-value | $h^2$ | p-value for $h^2$ |
|---|---|---|---|---|---|---|
| CE(20:3) | 0.21 | 0.09 | 2.34 | 2.31E-02 | 0.31 | 0.30 |
| LPC(15:0) | −0.22 | 0.09 | −2.54 | 1.39E-02 | 6.51E-16 | 1 |
| LPC(16:0) | −0.27 | 0.09 | −3.12 | 2.90E-03 | 3.82E-14 | 1 |
| LPC(17:0) | −0.21 | 0.09 | −2.34 | 2.30E-02 | 2.82E-14 | 1 |
| LPC(18:1e) | −0.21 | 0.09 | −2.44 | 1.81E-02 | 3.52E-17 | 1 |
| LPC(26:0) | −0.27 | 0.09 | −3.10 | 3.07E-03 | 0.056 | 0.87 |
| PC(39:3) | 0.18 | 0.09 | 2.12 | 3.84E-02 | 0.39 | 0.14 |
| TG(18:1_17:1_22:6) | −0.18 | 0.09 | −2.05 | 4.51E-02 | 0.31 | 0.05 |
| TG(18:1_18:1_22:5) | −0.23 | 0.09 | −2.69 | 9.58E-03 | 3.42E-15 | 1 |
| TG(18:1_20:4_22:6) | −0.21 | 0.09 | −2.41 | 1.96E-02 | 2.98E-15 | 1 |
| TG(19:0_18:1_18:1) | −0.18 | 0.09 | −2.14 | 3.73E-02 | 0.312 | 0.29 |
| GroupLPC | −0.24 | 0.09 | −2.74 | 8.32E-03 | 1.88E-15 | 1 |

Notes. Associations of GWAM with lipid residuals (adjusted for age, sex, education, BMI, lipid lowering medication, smoking status, experimental batch and *APOE* ε4 carrier status).

individuals, environmental factors were predominant, with only 13.3% of individual lipids analysed being significantly heritable, belonging mainly to the TG, DG and Cer lipid classes. We identified a higher number of gene-transcript associations with heritable as opposed to non-heritable lipids, and revealed unexpected biological roles of these lipids based on these transcript associations. Additionally, we found a small subset of non-heritable lipids to be associated with GWAM, suggesting a potential mechanism by which environmental influences can be conveyed on the lipidome. This powerful combination of lipidomics, transcriptomics and DNA methylation data in a twin study is the first of its kind and enables unique insight into lipidome heritability in older aged individuals.

## Heritability estimates

Heritable lipids had a moderate level of heritability (median $h^2$ = 0.433) which compares well with an estimate of 35.4% provided for metabolites from a genome-wide genotyping study in subjects aged 60 years and over (*Darst et al., 2019*), and another estimate of 0.37 from a family based heritability study (*Bellis et al., 2014*). Traditional lipid measures of LDL-C, HDL-C, total cholesterol and TG were significantly heritable, consistent with previous studies (*Liu et al., 2018*; *Goode et al., 2007*; *Mahaney et al., 1995*), though our estimates for these traits (range 0.40–0.47) were lower than estimates from other studies, reported to exceed 0.60 (*Liu et al., 2018*; *Beekman et al., 2002*), likely due to age differences. Interestingly, one of these studies found heritability estimates of these traits among Australian twins to be lower than the same estimates in Dutch and Swedish twin pairs (*Beekman et al., 2002*) (twin cohorts differing by age range), highlighting that differences in ethnicity, cohort and age can lead to substantial variance in reported heritabilities from study to study. Additionally, the substantial shared environment (C) component for HDL-C (0.27) is consistent with previous studies (*Liu et al., 2018*; *Mahaney et al., 1995*) that indicate that shared environment early in life is an important contributor to HDL-C variance later in life.

Comparing heritability at the level of individual lipid species, one recently published German twin study using data from *NutriGenomic Analysis in Twins* (NUGAT) yielded a similar range of heritabilities of 0–62% (*Frahnow et al., 2017*), finding 19 of 150 plasma lipid species to be highly heritable ($h^2$ > 0.40), not dissimilar to the proportion identified in the present study (27 of 207). However, the heritability of various classes often did not corroborate our findings. For example, NUGAT reported LPC and PE to be moderately heritable (0.25<$h^2$<0.35), while SMs had high heritability, as opposed to ceramides which were reported to be lowly heritable. By contrast, our study found high heritability of ceramides, and no significantly heritable SMs and virtually zero heritability of LPCs. A more recent publication of a Finnish population based study (FINRISK) reported SNP-based heritability of lipid species to be in the range 0.10–0.54 (*Tabassum et al., 2019*), and found Cer to be the most heritable species, corroborating findings from the present study and others (*McGurk et al., 2017*; *McGurk et al., 2019*), though heritability of some other lipid classes, such as LPC was markedly higher than reported in the present study. We have summarised our findings against the backdrop of previous literature in *Table 5*. One possible explanation for reported differences is that heritabilities may change across the lifespan (*Goode et al., 2007*). Age-dependent increases in heritability have been reported for LDL-C and HDL-C (*Goode et al., 2007*), while heritability estimates for BMI are lower in older adults compared to young adults (*Silventoinen et al., 2017*). In our cohort, we did find age-dependent heritability effects in both directions (*Supplementary file 2C*). This could be amplified when considering that the age range of NUGAT participants was 18 to 70 years, and 25 to 74 years for FINRISK whereas OATS consisted of much older individuals ranging from 69 to 93 years. Further, the age range may exacerbate the potential impact of pre- and post-menopausal status on lipid profiles (*Anagnostis et al., 2015*; *Saha et al., 2013*) among women, who comprise a majority of participants in both OATS (n = 179, 68.8%) and the NUGAT study (n = 58, 63%), though we only found minor evidence of sex-dependent heritability effects in our ageing cohort. In summary, while our heritabilis ranges were largely consistent with previous studies, there is some variance in heritability of particular lipid classes and individual species.

## Lipid associations with DNA methylation

To assess possible mechanisms contributing to variance of non-heritable lipids, we compared average DNA methylation levels over 450,000 different DNA methylation sites among MZ twins. DNA methylation is a well characterised epigenetic mechanism by which a gene expression profile can be

**Table 5.** Comparison of heritability estimates for traditional lipids and specific lipid classes/species summarising the current work and other published studies.

| Study and cohort details | Findings | Reference |
|---|---|---|
| Traditional lipids | | |
| Present study<br>75 MZ pairs, 55 DZ pairs<br>69–93 years | Range $h^2$: 0.404–0.427<br>HDL-C had substantial C-component 0.27 | |
| Qingdao Twin Registry<br>382 MZ pairs and 139 DZ pairs, mean age 51 ± 7 | Total Cholesterol and LDL-C 0.614, 0.655<br>HDL-C $h^2$ = 0.26, C-component = 0.478 | *Liu et al., 2018* |
| National Heart Lung and Blood Institute Veteran Twin Study;<br>235 MZ, 260 DZ pairs<br>48–63 years | Longitudinal increases in heritability across three time pts<br>Total Cholesterol (from 0.46 to 0.57), LDL-C (from 0.49 to 0.64), and HDL-C (from 0.50 to 0.62)<br>TG: $h^2$ = 0.40 | *Goode et al., 2007* |
| San Antonio Family Heart Study<br>N = 569, mean age 39.4 years | $h^2$HDL-C = 0.55, $h^2$TG = 0.53 | *Goode et al., 2007*;<br>*Mahaney et al., 1995* |
| Lipid Species/Classes | | |
| Present study | Range $h^2$: 0–0.59<br>Range heritable lipids: 0.287–0.59, median: 0.433<br>Heritable lipids: some Cer, TG, DG. Fewer PE, PC<br>Non-heritable lipids: LPC, SM, PI, CE | |
| Wisconsin Registry for Alzheimer's Prevention n = 1212, mean age 60.8 | Range 0.2–84.9%, median $h^2$ = 0.354<br>Median $h^2$ceramides = 0.48<br>Median $h^2$DG = 0.38 | *Darst et al., 2019* |
| San Antonio Family Heart study, n = 1212 mean age 39.52 | $h^2$range = 0.09–0.60<br>Median = 0.37<br>Heritable: almost all lipids, including Cer, TG, DG<br>Least heritable: LPC, alkyl-PE | *Bellis et al., 2014* |
| NUGAT Twin Study<br>34 MZ, 12 DZ twin pairs<br>18–70 years, median age 25 | Range $h^2$: 0–0.62 (19/152 lipids had $h^2$ > 0.40)<br>Heritable lipids: LPC, PE, SM<br>Non-heritable: Cer | *Frahnow et al., 2017*;<br>*Tabassum et al., 2019* |
| FINRISK n = 2181<br>25–74 years<br>SNP based heritability | SNP based range $h^2$: 0.10–0.54<br>Heritable lipids: Cer, LPC, SM, TG<br>Non-heritable: PI | *Tabassum et al., 2019* |
| n = 203 plasma samples from 31 families | Cer heritability range: 0.10–0.63 | *McGurk et al., 2017* |
| n = 999, 196 British families, mean age 45 | SNP-based Cer heritability range: 0.18–0.87 | *McGurk et al., 2019* |

regulated and inherited independent of the genetic sequence (*Egger et al., 2004*), and involves the addition of a methyl group (-CH$_3$) to the base cytosine of 5'-cytosine-phosphate-guanine-3' (CpG) dinucleotides (*Bird, 2002*; *Lim and Maher, 2010*). Methylation of CpG clusters around promoter regions of genes typically leads to suppression of gene transcription. Of salience, five LPCs and their summed total, which were extremely non-heritable (near zero), were significantly associated with GWAM. Although only a small subset of lipids showed significant associations with GWAM (just eight individual lipids of 180 non-heritable lipids), these findings do suggest that epigenetic factors such as DNA methylation could explain some of the variation associated with non-heritable lipids, especially very lowly heritable phospholipids and LPC, the least heritable lipid class in our data-set.

In previously published work, DNA methylation has been associated with environmental changes in lipid levels. Maternal lipids, passing from mother to child in utero at 26 weeks of gestation, lead to DNA methylation changes in the newborn (*Tindula et al., 2019*). The lipids associated with DNA methylation changes included phosphatidylcholine and lysolipids – phospholipid degradation products and choline could be an important precursor for DNA methylation. Similarly to our study, higher lipid metabolites were associated with lower methylation levels of genes involved in prenatal development. While the association of LPCs with DNA methylation has not previously been identified, it is worth noting that LPCs are a major source of polyunsaturated fatty acid (PUFA) for the brain (*Yalagala et al., 2019*) and regulate gene transcription through sterol regulatory-element binding protein (SREBP) pathways (*Chan et al., 2018*). Thus, LPC is an important lipid to convey dietary sources of PUFAs into the brain and regulate gene transcription.

These findings add to studies conducted in animal models which also show that nutrients taken by the mother are passed on to offspring during pregnancy, and may have a lasting impact on gene expression through DNA methylation (*Hoile et al., 2013*). Dietary restriction has also been shown to attenuate age-related hypomethylation of DNA in the liver, resulting in the downregulation of genes involved in lipogenesis and elongation of fatty acid chains in TGs, leading to a shift in the TG pool from long chain to medium and shorter chain TGs (*Hahn et al., 2017*). In summary, there is evidence to suggest that lipids can influence DNA methylation levels, while genes related to lipid metabolism can also be regulated in response to DNA methylation.

Interestingly, when we attempted to focus on DNA methylation at specific CpG sites within close proximity to genes whose transcripts were significantly associated with lipids, we found a few associations with lipids and with gene expression, but little overall evidence to indicate that DNA methylation drives gene expression of these transcripts. More work needs to be done to clarify these relationships using a larger sample size.

## Genetic correlations

High within-class genetic correlations between individual Cer, TG, and DG species (all r > 0.70) suggest similar genetic influences between lipids of the same class. Further, Cer species and monounsaturated SM also exhibited high genetic correlations, as did TG and DG. Metabolically, Cer and SM belong to the sphingolipid class where SM can be converted to Cer via sphingomyelin phosphodiesterase (*Pralhada Rao et al., 2013*), while TG and DG are interconvertible, where TG can be metabolised to DG by adipose triglyceride lipase (ATGL), or DG to TG through the addition of acyl CoA via DG acyltransferase (DGAT) (*Liang and Nishino, 2010*). Our results suggest that the heritable lipidome is regulated by overlapping genes which are associated with multiple lipids, especially lipids that belong to the same class, or are related by a connected metabolic pathway. Nevertheless, environmental correlations were still high for these lipids suggesting the importance of environmental factors on lipid levels. Traditional lipids (total triglyceride, LDL-C, HDL-C and total cholesterol) had low genetic and phenotypic correlations with individual lipid species, except for triglyceride measures, which were highly correlated with TG and DG species. This finding confirms previous results (*Tabassum et al., 2019*) and suggests some differences between variance in traditional lipid measures and variance in the lipidome at the individual lipid species level.

## Lipid-Transcriptome associations

Transcriptome associations of both heritable and non-heritable triglycerides, which represented the largest component of our lipidomics dataset, were assessed. We anticipated that both heritable and non-heritable lipids would have gene transcript probe associations, since endogenous triglycerides are derived from essential dietary fatty acids, such as linoleic acid, or other fatty acids substantially derived from dietary sources (such as linolenic acid and docosahexaenoic acid). Gut microbiota (microbiome) can also have an effect on the dietary lipidome, prior to absorption, representing another 'environmental' contributor, to lipid abundance and structure (*Just et al., 2018*). Although we hypothesized heritable lipids would be associated with gene transcripts involved in structural remodeling or transport of lipids (e.g. elongases, desaturases, synthases and synthetases), these transcripts were not represented in our lipid-blood transcriptome analysis. From this, we infer that the genes which are thought to account for the substantially heritable phenotype of our triglyceride group (i.e. via lipid metabolic processes) are not necessarily the same as those reflected in the lipid-transcriptome associations. This might be the case if the heritable aspect of our lipid list is driven by lipid modifying genes (such as desaturases, elongases, fatty acid synthases and synthetases), while the blood transcriptome is associated with the endogenous lipidome, which is a product of both environment and genetics (a feedback loop of sorts). We model this hypothesis in *Figure 4*. This is to say the blood transcriptome is malleable to both genetic and environmental influences and complements our finding that variance in lipid levels due to heritability is only partially accounted for by gene expression of associated transcripts.

## Biological effects of the lipid associated blood transcriptome

Our lipid-transcriptome analysis revealed strong associations of lipids with gene transcripts involved in modulating immune and vascular function. Interestingly, a previous twin study found a minor

subset of the immune system is modulated by genetic influences, such as the homeostatic cytokine response (*Brodin et al., 2015*), and many of the associated gene transcripts in the current study including Solute carrier family 45 member 3 (*SLC45A3*), *CPA3* and *HDC* were previously reported in a study of lipid and immune response (*Inouye et al., 2010*). Thus, some of the transcriptome associations uncovered could reflect lipid-modulated innate immune responses. Since this protein coding transcriptome has largely negative associations with lipid levels, we infer that it is moderating/suppressing inflammation or adverse vascular events. On the other hand, high fat diet in mouse models leads to elevated gene transcription related to white adipose tissue and liver metabolism, and after a prolonged high fat dietary regimen, activation of inflammatory pathways (*Liang et al., 2013*). We postulate that lipid levels are normally linked to the suppression of inflammatory responses to maintain homeostasis, but become associated with activation of inflammatory responses following metabolic overload, such as in diabetes mellitus or obesity (*Feng et al., 2016*; *Hubler and Kennedy, 2016*). Indeed, the authors of this study only noted significant upregulation of genes associated with inflammatory pathways after six weeks of high fat diet consumption, in contrast to genes associated with lipid metabolism, which were upregulated directly following a high fat diet.Since most of the associated lipid-protein coding transcriptome were membrane proteins, this suggests a possible interaction between lipids and protein function at the cellular surface. For example, vesicle associated membrane protein 8 (VAMP8) is involved in cellular fusion and autophagy. This would also explain transcripts being associated with proteins involved in phosphorylation and other signalling pathways. Altogether, the lipid-blood transcriptome associations indicate likely roles of lipids in inflammation, immune response, membrane and cell surface signalling as opposed to lipid metabolism.

## Limitations and future perspectives

There are some important limitations to this work. Firstly, this study covers a fairly wide age range in older aged adults (69–93 years). Very few heritability studies have focused on the lipidome in this age bracket. It is thereby important to stress that the findings of this study may not necessarily generalise to the whole population. We suspect that in our older cohort, environmental factors would dominate given the time in which these exposures are allowed to accumulate and shape the lipidome. Some of the heritabilities reported may vary longitudinally, owing to the dynamic contribution of genetic and environmental factors, and their interaction, across the lifespan (*Steves et al., 2012*). In particular, heritability estimates may decrease where unique environmental exposures accumulate with time and become a dominant force in lipid modulation. By contrast, heritabilities may also increase where certain genes become more active in older age to shape a given phenotype, potentially relating to lipids that may convey protective effects with ageing, as opposed to harmful effects (*Marenberg et al., 1994*). Given the age range of the cohort used, the results from the present study likely reflect a combination of both genetic and environmental influences on variation in the lipidome relevant to older age, and may provide important clues as to lipids and genes important in longevity. Some of these influences may underlie metabolic and lipidomic signatures previously described in very old individuals (*Wong et al., 2019a*; *Montoliu et al., 2014*; *Clement et al., 2019*; *Armstrong et al., 2017*). It is also important to emphasise that heritability estimates only represent the relative contribution of genetic and environmental influences. A 'low heritability' score does not necessarily imply that there are no additive genetic effects, but rather that variation in the lipid profile among twins is largely mediated by the shared or unique environment. Further, we acknowledge that though we have included as many participants as possible from this study, there may be insufficient power to make substantive conclusions. Nevertheless, we believe our findings to be a good starting point for further investigation.

Transcriptomics data obtained through the Illumina microarray provides a broad overview of many potential gene transcript associations with measured lipids from the same individuals. However, these data were obtained using RNA from blood cells, which presents potential biases in the types of associations uncovered and could account for some of the immune regulatory genes uncovered. Nevertheless, given the strict cutoff p-value employed in the analyses, it is likely these associations reflect true roles of these lipids in immune function, and the genes we uncovered have previously been identified in other lipid-transcriptomic studies (*Inouye et al., 2010*). We must emphasise that the transcriptome is influenced by many independent factors up- and downstream. The relationship between genetic variance (heritability) and the transcriptome is not clearcut.

Nevertheless, we find some evidence that the transcriptome is linked to heritable plasma lipids and may explain a small proportion of their heritability. Additionally, while ceramides were the most heritable lipids, there were no significant gene expression associations with these lipids. This could be due to very low endogenous expression of ceramide synthases in leukocytes (*Levy and Futerman, 2010*), though this pattern may be different in tissues where the most abundant CerS, CerS2, is highly expressed (*Laviad et al., 2008*), such as in the kidney or liver (*Levy and Futerman, 2010*).

Another major limitation is the fact that only average levels of DNA methylation (i.e. GWAM) were considered when associating with lipids, rather than DNA methylation at specific sites. This approach was necessary in order to avoid multiple testing correction for over 450,000 CpG methylation sites. The result is that the associated lipids showed at best suggestive significant associations with DNA methylation. The associations that we did find were for non-heritable lipids only, especially the least heritable LPCs, and were largely inverse. It is likely that based on previous studies, more significant associations with DNA methylation sites could be determined using greater selectivity of methylation sites at certain genomic regions. Further, as our analysis only showed that a small subset of non-heritable lipids were associated with GWAM, there is a still a lot variation in the lipidome not accounted for. CpG site specific analysis for particular genes did not find a relationship between DNA methylation and gene expression of these transcripts, though this analysis may lack power to detect these relationships. Other epigenetic mechanisms such as histone modification and chromatin structural changes could be implicated in regulating lipid metabolism, but are beyond the scope of this study.

We also acknowledge that our lipidomic analyses was conducted in whole plasma as opposed to within lipoprotein fractions, which limits insight into biological properties of lipid species. For example, serum albumin contains free fatty acids, which were not analysed in the present study, and would require separate analysis through gas-chromatography mass spectrometry. Since free fatty acids incorporate into phospholipids and triglycerides, their function may reveal greater complexity in lipid regulation at the gene level than we have described. Nevertheless, plasma is a common biological matrix for lipidomic study and likely includes both free lipids and plasma bound components, and represents a good comparative source for further, more detailed studies. Additionally, we advocate for increased focus on assay of individual lipids. Despite some redundancy between traditional measurements and the individual species (e.g. total TG correlated well genetically and phenotypically with individual TGs), substantial variance in many lipid classes is not well represented using traditional measurements, and heritability of a lipid class may differ from heritabilities of individual lipids of that class, likely owing to the broad range of heritability estimates obtained, as previously reported (*Frahnow et al., 2017*).

In the context of guidelines for lipid health and therapeutic targets, individual lipids may strengthen genetic associations with lipid loci that could be useful for assessing cardiovascular disease risk (*Tabassum et al., 2019*). Non-heritability of some lipids and the malleability of the lipid-blood transcriptome also suggests these lipids could be amenable to modification therapeutically. While there is still a long way to go before analysis of individual lipids has direct clinical utility, our study presents a useful first step towards understanding how the broad lipidome is regulated, especially in older individuals, and their putative functions beyond that of traditional lipids.

## Conclusion

In our study of older Australian twins combining lipidomics, transcriptomics and DNA methylation data, a small subset of plasma lipids was heritable and included largely Cer, TG and DG species. Most phospholipids, especially LPCs, were not significantly heritable. Significantly heritable lipids exhibited high genetic correlations between individual Cer, TG and DG species, as well as between Cer and SM, and between DG and TG, indicating shared genetic influences between lipids of the same class or metabolic pathways. Heritable lipids, especially TGs and DGs, were associated with a greater degree of gene transcript probe associations relative to the non-heritable lipids, and these transcripts were related to immune function and cell signalling rather than lipid metabolism directly. Thus, genes not related to lipid metabolism may still be associated with plasma lipid levels. Finally, associations of genome-wide average DNA methylation with highly non-heritable lipids, especially LPCs, suggest a potential mechanism by which environmental influences on lipids are conveyed. Overall, this study shows that a vast majority of plasma lipids are controlled by the environment, and hence modifiable, with genetic control still a major contributor to Cer, DG and TG lipid levels.

Further, our study suggests a complex interaction between lipids, environment, DNA methylation and gene transcription.

# Materials and methods

## Key resources table

| Reagent type (species) or resource | Designation | Source or reference | Identifiers | Additional information |
|---|---|---|---|---|
| Biological sample (*Homo sapiens*) | Fasting human EDTA plasma OATS Wave 3 | Sachdev, P.S., et al (2011). Cognitive functioning in older twins: the Older Australian Twins Study. Australasian journal on ageing 30 Suppl 2, 17–23. | Subject Cohort used: Wave three from the Older Australian Twins Study (OATS) Age range: 69–93 years | Plasma used for lipidomics analysis. Cohort also has genetics (SNPs) data and gene methylation data |
| Chemical compound, drug | SPLASH Lipidomix Mass Spec Standard | Avanti (Alabaster, Alabama, United States) | SKU 330707-1EA | Stable isotope labelled internal lipid standards |
| Other | QExactive Plus mass spectrometer and associated software: Xcalibur (3.1.66.10) and MS Tune (2.8 SP1 Build 2806)) | Thermo Fischer Scientific (Waltham MA United States) | MSMS | Mass spectrometer and controller software |
| Other | DIONEX UltiMate 3000 LC System and associated Chromeleon software | Thermo Fischer Scientific (Waltham MA United States) | LC and controller software | The LC system is comprised of an RS pump, RS column compartment and RS autosampler |
| Software, algorithm | Lipidsearch software v4.2.2 | Thermo Fischer Scientific (Waltham MA United States) | ThermoFisher Scientific software | Lipid identification and peak area integration |
| Chemical compound, drug | Acteonitrile UN 1648 | Honeywell Burdick and Jackson | HPLC grade solvent CAS 75-05-08 | Solvent used for preparing LC-MS Buffers Country of manufacture: Korea |
| Chemical compound, drug | Ammonium formate | Honeywell Fluka | HPLC grade reagent CAS 540-69-2 | Reagent used for preparing LC-MS Buffers UNIVAR analytical reagent Country of manufacture: Germany |
| Chemical compound, drug | Formic Acid (99%) UN 1779 | AJAX Finechem (Nuplex Industries, Australia) | AR Grade CAS 64-18-6 | Solvent used for preparing LC-MS Buffers |
| Chemical compound, drug | Milli-Q IQ 7000 purified Water | Merck Millipore | Purity monitored to a minimum of 18 MΩ resistivity | Purified water for preparing buffers and general laboratory use |
| Chemical compound, drug | Isopropanol | Honeywell Burdick and Jackson Material No. 10626668 Manufactured: USA | LC-MS grade CAS 67-63-0 | Solvent used for preparing LC-MS It is important to use LC-MS grade isopropanol in buffer B, to maintain low background signal for LCMSMS |

*Continued on next page*

*Continued*

| Reagent type (species) or resource | Designation | Source or reference | Identifiers | Additional information |
|---|---|---|---|---|
| Other | Acquity LC column LC-MS reverse phase column | Waters Corporation | Acquity UPLC CSH C18, 1.7 mm, 2.1 × 100 mm column SKU 186005297 | Includes Vanguard pre-column attachment. |
| Chemical compound, drug | Butanol for lipid extraction | Asia Pacific Specialty Chemicals, Thermo Fisher Scientific | CAS 71-36-3 | Extraction described: https://doi.org/www.frontiersin.org/articles/10.3389/fneur.2019.00879/full https://www.mdpi.com/2218-1989/5/2/389 |
| Chemical compound, drug | Methanol HPLC grade solvent for lipid extraction | AJAX Finechem (Nuplex Industries, Australia) | CAS 67-56-1 | |
| Commercial assay or kit | PAXgene blood RNA system | PreAnalytix, Qiagen | CAS 762165 CAS 762164 | RNA blood tube and extraction kit. Used as per manufacturer's protocol |
| Commercial assay or kit | Agilent Technologies 2100 Bioanalyzer | Agilent | G2939BA | RNA integrity number (RIN) assessment |
| Commercial assay or kit | Illumina Whole-Genome Gene Expression direct Hybridization Assay System HumanHT-12 v4 | Illumina, San Diego, CA | BD-103–0604 | Used as per manufacturer's protocol BD-901–1002 |
| Commercial assay or kit | Illumina Infinium HumanMethylation 450 BeadChip | Illumina, San Diego, CA | WG-314–1002 | Used as per manufacturer's protocol |
| Other | Beckman LX20 Analyser (clinical chemistry analysis of LDL-C, HDL-C, triglyerides) | Beckman Coulter, Australia | | Done at Prince of Wales hospital, Sydney. Timed endpoint method used for calculation of LDL-C. |
| Commercial assay or kit | *APOE* genotyping: Taqman genotyping assays *Assays:* C__3084793_20 (*rs429358*) & C_904973_10 (*rs7412*) | Poljak, A., et al. The Relationship Between Plasma Abeta Levels, Cognitive Function and Brain Volumetrics: Sydney Memory and Ageing Study. Curr Alzheimer Res 2016;13:243–55 | Applied Biosystems Inc, Foster city, CA | |
| Software, algorithm | ROpenMx 2.12.2 | Neale, M.C., et al. (2016). OpenMx 2.0: Extended Structural Equation and Statistical Modeling. Psychometrika 81, 535–549. | | SEM heritability analysis R package |

*Continued on next page*

*Continued*

| Reagent type (species) or resource | Designation | Source or reference | Identifiers | Additional information |
|---|---|---|---|---|
| Software, algorithm | Other R packages: Minfi, RNOmni, nlme, rcompanion, caret | Aryee MJ, et al. Minfi: a flexible and comprehensive bioconductor package for the analysis of Infinium DNA methylation microarrays. Bioinformatics 30(10), 1363–1369 (2014). *McCaw, 2019*. RNOmni: Rank Normal Transformation Omnibus Test. In. (R package *Pinheiro et al., 2019*. nlme: Linear and Nonlinear Mixed Effects Models. In. (R package *Mangiafico, 2019*. rcompanion: Functions to Support Extension Education Program Evaluation. In. (R package Kuhn, M., et al. (2018). caret: Classification and Regression Training. In. (R package | | |

## Cohorts

The study sample comprised participants aged between 69–93 years enrolled in the Older Australian Twin Study (OATS), established in 2007. The study recruited participants from three states in eastern Australia (QLD, NSW and VIC). The OATS collection included; patient data, including blood chemistry, MRI, neuropsychiatric assessment/cognitive tests, and medical exams performed over several visits (waves), each taken at an interval of 16–18 months, with the first visit denoted as 'Wave 1', second visit denoted as 'Wave 2' and so on. From OATS, we selected $n = 330$ participants who had available plasma from Wave 3; plasma from this wave collected within a period of up to 3 years apart. Of these, 260 participants were eligible for heritability analyses, including 150 monozygotic twins (75 pairs in total; 25 male, 50 female), and 110 dizygotic twins (55 pairs in total; 31 males, and 79 females). The study protocol for OATS has been previously published (*Sachdev et al., 2009*; *Sachdev et al., 2013*; *Sachdev et al., 2011*). Participants who had significant neuropsychiatric disorders, cancer, or life threatening illness were excluded from this study.

## Plasma collection, handling and storage

Blood collection, processing and storage were performed under strict conditions to minimize pre-analytical variability (*Wong et al., 2017*). Fasting EDTA plasma was separated from whole blood within 2–4 hr of venepuncture and immediately stored at −80℃ prior to bio-banking. Samples then underwent a single freeze thaw cycle for the purpose of creating aliquots, which minimizes subsequent freeze thaw cycles for specific experiments. EDTA plasma was chosen as the anticoagulant since it chelates divalent metals, thereby protecting plasma constituents from oxidation, which is particularly important for lipids. Thereafter, lipid extractions were performed within 15 min of freeze thawing and extracts stored at −80℃ and analysed within two months of extraction.

## Targeted assays of plasma lipids

Plasma total cholesterol, LDL-C, HDL-C and TG were measured by enzymatic assay at SEALS pathology (Prince of Wales Hospital) as previously described (*Song et al., 2012*), using a Beckman LX20 Analyzer with a timed-endpoint method (Fullerton, CA). LDL-C was estimated using the Friedewald equation (LDL-C = total cholesterol - HDL-C - triglycerides/2.2).

## APOE genotyping

DNA was extracted from samples using established procedures (*Muenchhoff et al., 2017*). Genotyping of two *APOE* single nucleotide polymorphisms (SNPs rs7412, rs429358) was performed using Taqman genotyping assays (Applied Biosystems Inc, Foster City, CA) to determine the *APOE* haplotype, which has three alleles (ε2, ε3, ε4).

## Lipid extraction from plasma: Single phase 1-butanol/methanol

Lipid internal standards (SPLASH Lipidomix Mass Spec Standard) were purchased from Avanti (Alabaster, Alabama, United States) and diluted ten-fold in 1-butanol/methanol (1:1 v/v). Plasma extraction was performed in accordance with a single phase extraction as previously described (*Alshehry et al., 2015*; *Wong et al., 2019b*). Briefly, we added 10 µL of 1:10 diluted SPLASH internal lipid standards mixture to 10 µL plasma in Eppendorf 0.5 mL tubes. 100 µL of 1-butanol/methanol (1:1 v/v) containing 5 mM ammonium formate was then added to the sample. Afterwards, samples were vortexed for 10 s, then sonicated for one hour. Tubes were centrifuged at 13,000 g for 10 min. The supernatant was then removed via a 200 µl gel-tipped pipette into a fresh Eppendorf tube. A further 100 µl of 1-butanol/methanol (1:1 v/v) was added to the pellet to re- extract any remaining lipids. The combined supernatant was dried by vacuum centrifugation and resuspended in 100 µl of 1-butanol/methanol (1:1 v/v) containing 5 mM ammonium formate and transferred into 300 µl Chromacol autosampler vials containing a glass insert. Samples were stored at −80˚ C prior to LC-MS analysis. The robustness and reproducibility of this extraction method has been previously demonstrated (*Wong et al., 2019b*) in our laboratory, with variation in human plasma ranges of measurement between individuals across age, sex (*Wong et al., 2019a*) and by *APOE* genotype (*Wong et al., 2019c*) reported.

## Liquid chromatography/Mass spectrometry

Lipid analysis was performed by LC ESI-MS/MS using a Thermo QExactive Plus Orbitrap mass spectrometer (Bremen, Germany) in two experimental batches separated by a month. A Waters ACQUITY UPLC CSHTM C18 1.7 um, 2.1 × 100 mm column was used for liquid chromatography at a flow rate of 260 µL/min, using the following gradient condition: 32% solvent B to 100% over 25 min, a return to 32% B and finally 32% B equilibration for 5 min prior to the next injection. Solvents A and B consisted of acetonitrile:MilliQ water (6:4 v/v) and isopropanol:acetonitrile (9:1 v/v) respectively, both containing 10 mM ammonium formate and 0.1% formic acid. Product ion scanning was performed in positive ion mode. Sampling order was randomised prior to analysis.

## Lipidsearch v4.2.2 search parameters

Lipidsearch software v4.2.2 (Thermo Fischer Scientific, Waltham MA) was applied to perform searches on raw files using the databases 'General' and 'labelled standards'. For peak detection, recalc isotope was set to 'ON', RT interval = 0.0 min. We used product search for LC-MS method and the precursor and product tolerances were set at 5.0 ppm and 8.0 ppm respectively. The intensity threshold was 1% parent ion, and the m-score threshold was set to 2.0. For quantitation, mz tolerance was set at −5.0 ppm to 5.0 ppm, and the retention time range was set at −0.5 to 0.5 min. The m-score threshold was 5.0, and all lipid classes were selected for inclusion. Ion adducts included +H, +NH4 for positive ion mode.

## Alignment and peak detection/analysis

The raw data were aligned, chromatographic peaks selected, specific lipids identified and their peak areas integrated using LipidSearch. Owing to the large number of RAW files being processed, the alignment step was performed in four separate batches, with a maximum of 100 samples aligned at any one time, and the data collated and exported to an Excel spreadsheet for manual processing and statistical analysis. Only lipids that were present in all four alignment batches were included in our analysis. The raw abundances (peak areas) were normalised by dividing each peak area by the raw abundance of the corresponding internal standard for that lipid class; for example, all phosphatidylcholines were normalised using 15:0-18:1(d7) PC. The intra-assay coefficient of variation (CV) was calculated by dividing the standard deviation of the normalised abundances by the mean across lipid species. Lipid ion identifications were filtered using the LipidSearch parameters rej = 0 and average

peak quality >0.75. Furthermore, identifications with CV < 0.4 from repeated injections of quality control standards every 20 runs were included. These controls included: (i) blank, to check for column and chromatography background levels, (ii) internal standards only, to check on system performance across a long sequence of runs, (iii) quality control plasma, to check for between run performance and enable calculation of between run and within run assay CV%. Where duplicate identifications were found on LipidSearch (i.e. lipid IDs with identical m/z and annotations, and similar retention times), the lipid ID with the lowest CV%, and highest peak quality score was used. When necessary, the average m-score (match score, based on number of matches with product ion peaks in the spectrum [20]) was also used to differentiate closely related lipid species, with the lipid having the highest m-score selected. All other duplicates were excluded from analysis. Lipid group-sums were produced by adding lipids within a defined class/subclass together, such as total mono-unsaturated triglycerides (TG), total ceramides (Cer) etc.

## Microarray gene expression

Fasting blood samples for gene expression analyses were collected. The methods for gene expression data collection analyses have previously been described (*Ciobanu et al., 2018*). Briefly, PAXgene Blood RNA System (PreAnalytiX, QIAGEN) was used to extract total RNA from whole blood collected in PAXgene tubes following overnight fasting. RNA samples with RNA integrity number (RIN) ≥6 as measured by the Agilent Technologies 2100 Bioanalyzer were used in subsequent analyses (*Gallego Romero et al., 2014*). Assays for gene expression were performed using the Illumina Whole-Genome Gene Expression Direct Hybridization Assay System HumanHT-12 v4 (Illumina Inc, San Diego, CA, USA) in accordance with standard manufacturer protocols. Quality control (QC) and pre-processing of raw gene expression intensity values extracted from GenomeStudio (Illumina) were performed using the R Bioconductor package limma (*Ritchie et al., 2015*). Background correction and quantile normalisation was done using the neqc function. Expressed probes with detection p-value<=0.05 were retained for analysis. After pre-processing and filtering, 308 samples and 36,053 transcripts were available for gene expression analysis. After overlapping with the lipids data 290 samples were available for lipids – gene expression analysis. Gene abbreviations used in the text are based on Gene Ontology nomenclature.

## DNA methylation

Genome-wide DNA methylation data for 113 monozygotic twin pairs was generated using an established genomics provider using peripheral blood DNA collected at baseline (*Armstrong et al., 2017*). Randomisation of co-twins across the arrays was performed within experiments. DNA methylation status was assessed using the Illumina Infinium HumanMethylation450 BeadChip (Illumina Inc, San Diego, CA, USA). Background correction was applied to raw intensity data and the R *minfi* package was used to generate methylation beta values (ranging from 0 to 1) (*Aryee et al., 2014*). Quantile normalisation was used. We excluded sex chromosome probes, probes containing SNPs, cross-reactive probes as well as probes not detected in all samples from analysis (*Chen et al., 2013*). Following these quality control (QC) procedures, 420,982 out of 485,512 probes remained. White blood cell composition was estimated using a previously described method (*Houseman et al., 2012*), implemented in *minfi*. After filtering methylation outliers using the preprocessQantile function of the minfi package with default parameters, out of the 217 samples with methylation data, 135 overlapped with lipids data. Genome wide Average Methylation (GWAM) for each sample across all the probe level beta values were calculated.

## Data analysis

### Data transformations

Since different sets of covariates are used to adjust for the lipid levels, gene expression and methylation, we have first obtained residuals after adjusting for standard confounders in order to obtain lipid and gene expression profiles independent of cohort characteristics. Residuals for lipids were obtained after adjusting for age, sex, education, BMI, lipid lowering medication, smoking status, experimental batch and *APOE* ε4 carrier status, which were then inverse normal transformed using the R package RNOmni (*McCaw, 2019*). This transformation eliminated experimental batch separation effects (*Figure 2—figure supplement 1A and B*). Residuals for gene expression were obtained

after adjusting for age, sex, experimental batch, RIN, blood cell counts (eosinophils, lymphocytes, basophils and neutrophils - obtained using standard laboratory procedures by Prince of Wales SEALS Pathology). Residuals for methylation beta values were obtained after adjusting for age, sex, BMI and estimated white blood cell counts (CD8T, CD4T, NK, B-cell, monocytes, and granulocytes). Residuals were used for all the analyses presented here.

## Heritability estimation

Heritability was estimated using SEM. Under the SEM the phenotypic covariance between the twin pairs is modelled as a function of additive genetic (A), shared environmental (C) and unique environmental (E) components. In the narrow sense heritability is defined as the ratio of additive genetic variance [Var(A)] to the total phenotypic variance [Var(A)+Var(C)+Var(E)]. The model containing these three parameters (A, C and E) is known as the ACE model. For model parsimony and test concerning the variance parameters, models with only A and E components, known as AE model, and the models with CE and E components would be fit and compared with the full ACE model (*Neale and Cardon, 1992*). Genetic and environmental correlations were estimated using the bivariate Cholesky model. Heritability, genetic correlations and environmental correlations under the twin SEM were estimated using the R OpenMx (2.12.1.) package (*Neale et al., 2016*).

## Sex- and age-related differences in heritability

Differences in heritability between the two sexes were examined using the sex heterogeneity model. This model is similar to the general ACE model but additional parameters are used to represent the genetic and environmental effects of male and female samples. The male and female path coefficients are used to model the opposite sex pair in DZ twins. The full likelihood is the sum of likelihoods under MZ, DZ and opposite sex pairs. Test of equality of the parameters under male and female samples were examined by likelihood ratio test comparing the heterogeneity model against a constrained homogeneity model assuming the same set of parameters for male and female samples. Age-related change in heritability was examined using a gene-environment interaction model (*Purcell, 2002*). The path coefficients under this model were further decomposed to accommodate the age effects.

## Association tests

Test of association of lipids with probe level gene expression were performed using the linear mixed model and the lme function in R package nlme (*Pinheiro et al., 2019*). Gene expression and lipid residuals (adjusted for age, sex, education, BMI, lipid lowering medication, smoking status, experimental batch and *APOE* ε4 carrier status) were used as independent and dependent variables respectively in these models. A p-value threshold of $1.39 \times 10^{-6}$ (0.05/35971, obtained by Bonferroni conservative correction for total number of probes) was used to define significant associations of lipids with probe level gene expression.

Similarly, lipid residuals were used as dependent variable and average methylation value was used as the independent variable to test the association of lipids with methylation. The proportion of variance in lipids explained by the gene expression variation and methylation variance were estimated based on the log-likelihoods as implemented in the R package rcompanion (*Mangiafico, 2019*). For most of the lipids, multiple gene expression probes were associated. Hence to avoid overfitting and multi-collinearity, we used penalized regression methods as implemented in glmnet of the R package caret (*Kuhn et al., 2018*) to reduce the number of probes in the regression model. The list of probes retained in the glmnet model was used to estimate the variance contributed by the gene expression.

For analysis of GWAM (*Table 4*), $r^2$ is McFadden's pseudo-$r^2$. p-value for $h^2$ is the p-value for test of significant additive genetic effects ($h^2$ = heritability). Thus p-value for $h^2$ <0.05 indicates significant heritability. Regression coefficients are based on average methylation at CpG sites excluding any with known SNPs influencing lipid levels.

## Lipid shorthand notation

Lipids are named according to the LIPID MAPS convention (*Fahy et al., 2009*). Lipid abbreviations are as follows: ceramide (Cer), cholesterol ester (CE), diacylglycerol (DG), lysophosphatidylcholine

(PC), phosphatidylcholine (PC), phosphatidylethanolamine (PE), phosphatidylinositol (PI), sphingo-myelin (SM) and triglyceride (TG).

### R-Scripts

R code scripts for major analyses described in the Data Analysis are available by author request.

## Acknowledgements

We acknowledge the contribution of the OATS research team (https://cheba.unsw.edu.au/project/older-australian-twins-study) to this study. The OATS study has been funded by a National Health and Medical Research Council (NHMRC) and Australian Research Council (ARC) Strategic Award Grant of the Ageing Well, Ageing Productively Program (ID No. 401162) and NHMRC Project Grants (ID 1045325 and 1085606). We thank the Rebecca Cooper Medical Research Foundation for their research support. We thank the participants for their time and generosity in contributing to this research. We would also like to acknowledge Ms. Mahboobeh Housseini's assistance with OATS plasma biobanking.

This research was facilitated through Twins Research Australia, a national resource in part supported by a NHMRC Centre for Research Excellence Grant (ID: 1079102).

## Additional information

### Funding

| Funder | Grant reference number | Author |
|---|---|---|
| National Health and Medical Research Council | | Nady Braidy<br>Perminder S Sachdev |
| Australian Research Council | | Nady Braidy |
| Rebecca L. Cooper Medical Research Foundation | | Nady Braidy<br>Anne Poljak<br>Perminder S Sachdev |
| National Health and Medical Research Council | 1045325 | Perminder S Sachdev |
| National Health and Medical Research Council | 1085606 | Nady Braidy |

The funders had no role in study design, data collection and interpretation, or the decision to submit the work for publication.

### Author contributions

Matthew WK Wong, Conceptualization, Data curation, Formal analysis, Validation, Investigation, Visualization, Methodology, Writing - original draft, Writing - review and editing; Anbupalam Thalamuthu, Conceptualization, Resources, Data curation, Software, Formal analysis, Validation, Visualization, Methodology, Writing - original draft, Writing - review and editing; Nady Braidy, Conceptualization, Supervision, Funding acquisition, Project administration, Writing - review and editing; Karen A Mather, Formal analysis, Validation, Investigation, Writing - original draft, Writing - review and editing; Yue Liu, Liliana Ciobanu, Bernhardt T Baune, Investigation, Writing - review and editing; Nicola J Armstrong, Software, Formal analysis, Writing - review and editing; John Kwok, Formal analysis, Writing - review and editing; Peter Schofield, Margaret J Wright, David Ames, Teresa Lee, Resources, Project administration, Writing - review and editing; Russell Pickford, Resources, Data curation, Methodology, Writing - review and editing; Anne Poljak, Conceptualization, Formal analysis, Supervision, Investigation, Visualization, Writing - original draft, Project administration, Writing - review and editing; Perminder S Sachdev, Conceptualization, Resources, Supervision, Funding acquisition, Project administration, Writing - review and editing

## Author ORCIDs

Matthew WK Wong https://orcid.org/0000-0001-6913-4558
Nicola J Armstrong http://orcid.org/0000-0002-4477-293X
Margaret J Wright http://orcid.org/0000-0001-7133-4970
Perminder S Sachdev https://orcid.org/0000-0002-9595-3220

## Ethics

Human subjects: OATS was approved by the Ethics Committees of the University of New South Wales and the South Eastern Sydney Local Health District (ethics approval HC17414). All work involving human participants was performed in accordance with the principles of the Declaration of Helsinki of the World Medical Association. Informed consent was obtained from all participants and/or guardians.

## Decision letter and Author response

Decision letter https://doi.org/10.7554/eLife.58954.sa1
Author response https://doi.org/10.7554/eLife.58954.sa2

## Additional files

### Supplementary files

• Source code 1. R script for analyses.

• Supplementary file 1. (**A**) Heritable lipid species. Standardized additive genetic ($h^2$ = heritability), shared environment ($h^2_c$) and unique environment ($h^2_E$) variance components (95% CI) of lipids were obtained using the ACE model. The columns p-AE, p-CE, and p-E, respectively, denote the p-values from the likelihood ratio test comparing ACE model vs AE, CE, and E models. p-CE is also the p value for heritability because testing the component A = 0 is equivalent to testing that the heritability is zero. C.I. indicates confidence interval; DZ, dizygotic; ICC, intraclass correlation coefficient; MZ, monozygotic. (**B**) Heritability of summed lipid groups. Standardized additive genetic (A = heritability), shared environment (C) and unique environment (E) variance components (95% CI) of lipids were obtained using the ACE model. The columns p-AE, p-CE, and p-E, respectively, denote the p-values from the likelihood ratio test comparing ACE model vs AE, CE, and E models. p-CE is also the p value for heritability because testing the component A = 0 is equivalent to testing that the heritability is zero. C.I. indicates confidence interval; DZ, dizygotic; ICC, intraclass correlation coefficient; MZ, monozygotic. CL_TG49 and CL_TG62 represent sum of triglycerides with 44–49 total carbons, and 56–62 total carbons respectively while Cer(d18:1/X) represents sum of all ceramides with an 18:1 acyl chain in the sn-1 position.

• Supplementary file 2. (**A**) Full heritability list for all lipids. HA - Heritabiltiy; HC - Shared envrionmental component; HE - Unique environmental component; MZICC - Intraclass correlation in MZ twins; DZICC - Intraclass correlation in DZ twin; LT and UT denote lower and upper confidence limits; SignificantlyHeritable = 1 if declared heritable and 0 if not significantly heritable; SigProbAssoc = 1 if at least one gene expression probe is associated with the lipid; 0 if none is associated. (**B**) Sex heterogeneity test. Only significantly heritable lipids from (B) included. HA - Heritabiltiy; HC - Shared envrionmental component; HE - Unique environmental component; MZICC - Intraclass correlation in MZ twins; DZICC - Intraclass correlation in DZ twins; LT and UT denote lower and upper confidence limits; f denotes females, m denotes males; Hom_Pval is the p-value for the sex heterogenity test (p<0.05 indicates significant sex differences in heritability for that lipid); ICCmf - Intraclass correlation coefficient between opposite sex pairs. (**C**) ACE Age-related changes. Listed are heritability estimates for lipids at various ages (65 to 95). (**D**) Significant probe associations. Nprobes = Number of significantly associated gene expression probes with the lipid. NusedInModel = number of probes used for the calculation of variance explained after fitting the penalised regression model. (**E**) Significant transcript association with lipids. (**F**) Lipid-gene table. Entry one in the table indicates a significantly associated gene probe in the column with the lipid in the row and 0 indicates no association. (**G**) Lipid-gene by saturation index. RowSum is the sum of lipids significantly associated with a probe in each saturation level. RowSum_sigherit is the sum of heritable lipids significantly associated with a

probe. Rowsum_nonsig is the sum of non-heritable lipids significantly associated with a probe. (**H**) Lipid-gene by total carbons. (**I**) Lipid expression associations with DNA methylation at CpG sites close to gene transcripts significantly associated with lipids. (**J**) Gene expression associations with DNA methylation at CpG sites close to gene transcripts significantly associated with lipids.

- Supplementary file 3. Lipid (triglyceride)-gene expression associations listed by heritability and degree of saturation.

- Transparent reporting form

### Data availability

All data generated or analysed during this study are included in the manuscript and supporting files.

The following datasets were generated:

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
