## [Decision Letter]

**Acceptance summary:**

This study represents a comprehensive analysis of the heritability of the aging plasma lipidome in healthy older twins enrolled in the Older Australian Twins Study (OATS). Exploration of gene expression and DNA methylation profiles showed heritability under the classical ACE twin model. Heritable triglycerides were associated with gene transcripts relating to immune and cell signaling molecules, whereas variability in non-inheritable lipids was accounted for by genome-wide average DNA methylation. In all, these studies reveal interacting genetic and environmental regulation of the plasma lipidome.

**Decision letter after peer review:**

Thank you for submitting your article "Genetic and environmental determinants of variation in the plasma lipidome of older Australian twins" for consideration by *eLife*. Your article has been reviewed by two peer reviewers, and the evaluation has been overseen by Mone Zaidi as Reviewing Editor and Matthias Barton as the Senior Editor. The reviewers have opted to remain anonymous.

The reviewers have discussed the reviews with one another and the Reviewing Editor has drafted this decision to help you prepare a revised submission.

Summary:

This is an extensive study of older Australian twins (OATS) reporting the heritability of numerous lipid species in plasma of a collection of 75 monozygotic and 55 dizygotic twins (aged 69 – 93 years). They examine lipidomics, transcriptomics and DNA methylation to determine if there are certain lipid profiles that appear inheritable and are driven by underlying RNA levels or DNA methylation. The major findings are that overall 3% of lipid species displayed significant heritability estimates, with the largest fractions being among TGs, ceramides, and diacylglycerols.

Essential revisions:

Overall the body of work is substantive and the data has been analyzed with sufficient statistical rigor. The authors rightly note in the manuscript the scientific shortcomings of their DNA methylation analysis. Four key issues need to be addressed.

1) The manuscript's major shortcoming is related to the analysis. This is a descriptive study that makes a long-winded cursory attempt at analysis in the Discussion. The manuscript should be revised to condense the analysis into clearer, bolder conclusions and placed into the context of prior studies. Perhaps a supplementary figure could be generated comparing the findings of the current study with those of prior studies so that commonalities and differences can be more readily appreciated by readers. For example there is a publication on heritability of ceramides and other lipids that should be referenced and included in the analysis: McGurk et al., 2019.

2) From a methodological standpoint, the authors may wish to acknowledge that the study population is probably not ideal for the research question. For many traits, heritability is greatest studied in younger individuals (e.g. PMID: 8127331), and this cohort of older twins may therefore significantly underestimate heritability. This is further suggested by the authors' finding of h^2^ for HDL-C of 0.419, which is much lower than that reported in other studies (typically 0.6 – 0.8). This seems a more likely explanation rather than this being a unique feature of Australian twins as the authors suggest.

3) A second methodological issue is that the lipidomic profiles were assessed in whole plasma, which provides comparatively less insight into the biological properties of these lipid species compared to approaches that assay lipid concentrations within relevant compartments (e.g. lipoproteins, RBCs, albumin etc). The authors should discuss this limitation and how it may influence their results. This is particularly important given that many of the lipid species assessed are highly correlated with traditional lipid values (e.g. between TG and DG), and therefore the question arises as to whether the lipidomic measurements are really just an indirect means of assessing traditional lipid and lipoprotein fractions.

4) Were there differences in heritability observed for male vs. female MZ twins?

---

## [Author Response]

Essential revisions:Overall the body of work is substantive and the data has been analyzed with sufficient statistical rigor. The authors rightly note in the manuscript the scientific shortcomings of their DNA methylation analysis. Four key issues need to be addressed.1) The manuscript's major shortcoming is related to the analysis. This is a descriptive study that makes a long-winded cursory attempt at analysis in the Discussion. The manuscript should be revised to condense the analysis into clearer, bolder conclusions and placed into the context of prior studies. Perhaps a supplementary figure could be generated comparing the findings of the current study with those of prior studies so that commonalities and differences can be more readily appreciated by readers. For example there is a publication on heritability of ceramides and other lipids that should be referenced and included in the analysis: McGurk et al., 2019.

We consolidated the Discussion, especially in relation to comparing heritability results against previous studies, placed an overarching summary at the beginning of the Discussion, and as suggested, a table (Table 5) has now been included to summarise this information. Some statements originally written under “lipid-transcriptome associations” of the Discussion have now been relocated into the Results section to make reading easier, and the Discussion more to-the-point.

2) From a methodological standpoint, the authors may wish to acknowledge that the study population is probably not ideal for the research question. For many traits, heritability is greatest studied in younger individuals (e.g. PMID: 8127331), and this cohort of older twins may therefore significantly underestimate heritability. This is further suggested by the authors' finding of h^2^ for HDL-C of 0.419, which is much lower than that reported in other studies (typically 0.6 – 0.8). This seems a more likely explanation rather than this being a unique feature of Australian twins as the authors suggest.

The reviewers point out an interesting observation re PMID: 8127331 and a possible reason why we have identified a lower HDL-C h^2^ value. However, we disagree “that the study population is probably not ideal for the research question”. The research question sought to address the heritability of plasma lipids in older age groups, so selecting an older age twin group to address this question was appropriate.

Nevertheless, we are aware that heritabilities can fluctuate as a function of age, which has been previously reported for traditional lipids. Though our study size was underpowered for a robust analysis of variation in heritability by age, however we explored this possibility, and have included an analysis (Supplementary file 2C), which supports a decrease of HDL with age, from 0.479 (age 65) to 0.38 (age 95).

Regarding PMID: 8127331, the “trait” studied was death from coronary artery disease in twins ranging between 36 – 75 years of age at baseline. The study did indeed demonstrate quite nicely that risk of death of a monozygotic twin was reduced, the older the age of death of the twin. It seems that early death from this kind of disease likely carries with it genetic risk factors. However, the trait we study is simply inheritance of specific lipid profiles – some of which may be advantageous to health, as well as the converse. We have no reason to expect that heritability decreases with age as a general rule. Actually, there is evidence to the contrary, as follows: The literature points out that there is high heritability of traits that lead to disease (see Steves et al., 2012). Since it would be a fair expectation that disease also leads to earlier death, then it should also not be too surprising that for lethal traits, heritability is reduced with age.

On the other hand, heritability could stay stable or even increase with age for salutary traits. For instance, heritability of intelligence/cognitive ability increases with age (see: Ploman and Deary Mol Psychiatry. 2015 Feb; 20(1): 98–108 , Haworth et al. Mol Psychiatry. 2010 Nov; 15(11): 1112–1120. The heritability of general cognitive ability increases linearly from childhood to young adulthood). Furthermore heritability of longevity increases with age, particularly after age 60 (see: Brooks-Wilson Hum Genet. 2013; 132(12): 1323–1338 Genetics of healthy aging and longevity, Hjelmborg et al. Genetic influence on human lifespan and longevity. Hum Genet. 2006;119:312–321).

These studies suggest that heritability seems to decrease with age for harmful traits but to increase with age for salutary traits. We anticipate a mix of age-trends for lipids, some disease associated while others are possibly associated with good health/longevity. To satisfy the reviewer, we have performed heritability analysis where h^2^ estimates were calculated at various (cross-sectional) ages within our cohort, and identified both increases and decreases in heritability with age of different lipids (Supplementary file 2C), supporting the reviewers’ suggestion of age related variation in heritability, though not necessarily in a downwards direction with age. This has been acknowledged in the “Limitations and Future Perspectives” section of the Discussion (first paragraph).

“In particular, heritability estimates may decrease where unique environmental exposures accumulate with time and become a dominant force in lipid modulation. By contrast, heritabilities may also increase where certain genes become more active in older age to shape a given phenotype, potentially relating to lipids that may convey protective effects with ageing, as opposed to harmful effects.”

3) A second methodological issue is that the lipidomic profiles were assessed in whole plasma, which provides comparatively less insight into the biological properties of these lipid species compared to approaches that assay lipid concentrations within relevant compartments (e.g. lipoproteins, RBCs, albumin etc). The authors should discuss this limitation and how it may influence their results. This is particularly important given that many of the lipid species assessed are highly correlated with traditional lipid values (eg between TG and DG), and therefore the question arises as to whether the lipidomic measurements are really just an indirect means of assessing traditional lipid and lipoprotein fractions.

We thank the reviewers for this suggestion. Our focus on plasma lipids was a limitation related to the additional amount and cost of processing and analysis required if compartmental experiments were performed considering the additional omics (transcriptomics and DNA methylation) data already presented within the paper, and also concern about the conclusions we could make based on our limited sample size. Even so, our extraction technique likely recovers free lipids as well as any bound within plasma components such as lipoproteins and albumin.

We provide our results as a good starting point that could lead towards further, detailed examination in future studies of larger cohort size. We have addressed this point within the Discussion:

“We also acknowledge that our lipidomic analyses was conducted in whole plasma as opposed to within lipoprotein fractions, which limits insight into biological properties of lipid species. […] Nevertheless, plasma is a common biological matrix for lipidomic study and likely includes both free lipids and plasma bound components, and represents a good comparative source for further, more detailed studies.”

As to the question of whether the lipidomic measurements are really just an indirect means of assessing traditional lipid and lipoprotein fractions, there is an expected degree of redundancy considering that total triglycerides are comprised of the sum of individual species. However we have noted here and in previous work that total cholesterol, LDL-C and HDL-C have only modest phenotypic correlations with individual lipid species, and levels of phospholipids and sphingolipids are not represented well by variance in total triglycerides. Thus there is a lot of variance in plasma lipids not explained by traditional lipids. Further, a recent study (Tabassum et al., 2019) using the FINRISK cohort showed that individual lipid species can strengthen genetic associations at known lipid loci and hence give more statistical power compared against traditional lipids with the same sample size. We have incorporated these considerations in the last two paragraphs of the Discussion.

4) Were there differences in heritability observed for male vs. female MZ twins?

To address this question, we ran a heritability analysis under a sex heterogeneity model. Here, four lipids had suggestive levels of significance (unadjusted p-value<0.05) to indicate differences in heritability estimates between male and female samples among significantly heritable lipids. However, due to the skewed female-male distribution, the sample size is not adequate to run a robust heterogeneity model. We have nevertheless included this information as Supplementary file 2B.